# Concept Probing: Where to Find Human-Defined Concepts

**Manuel de Sousa Ribeiro**                                      MAD.RIBEIRO@FCT.UNL.PT
**Afonso Leote**                                                      A.LEOTE@FCT.UNL.PT
**João Leite**                                                        JLEITE@FCT.UNL.PT
*NOVA LINCS, NOVA School of Science and Technology, NOVA University Lisbon, Portugal*

**Editors:** Leilani H. Gilpin, Eleonora Giunchiglia, Pascal Hitzler, and Emile van Krieken

## Abstract

Concept probing has recently gained popularity as a way for humans to peek into what is encoded within artificial neural networks. In concept probing, additional classifiers are trained to map the internal representations of a model into human-defined concepts of interest. However, the performance of these probes is highly dependent on the internal representations they probe from, making identifying the appropriate layer to probe an essential task. In this paper, we propose a method to automatically identify which layer's representations in a neural network model should be considered when probing for a given human-defined concept of interest, based on how informative and regular the representations are with respect to the concept. We validate our findings through an exhaustive empirical analysis over different neural network models and datasets.

## 1. Introduction

Artificial neural networks have been shown to achieve state-of-the-art results in a wide range of domains, playing a key role in addressing perceptual tasks (Hatcher and Yu, 2018). Despite their performance, neural networks remain largely opaque, due to their subsymbolic internal representations, which provide limited transparency regarding their decision-making process (Guidotti et al., 2019). This limitation has sparked renewed interest in Neuro-Symbolic AI (Besold et al., 2021), a field that seeks to bridge the gap between subsymbolic learning and symbolic abstraction, thereby leveraging the qualities of both approaches.

The increasing use of neural networks in sensitive domains, performing tasks that were once reserved for human judgment, and the need to leverage existing neural network models led to the research field of Explainable AI (Zhang et al., 2021) – a field that focuses on the development of methods to help improve a model's interpretability. Among the various approaches for interpreting existing neural networks, concept probing has emerged as a key methodology for understanding what they encode (Belinkov, 2022). The main idea is simple: for each human-defined concept of interest that one wants to probe, a model – referred to as a *probe* – is trained to map the internal representations of a neural network into the respective values of the concept of interest. After training, a probe can be used to observe the value of its concept of interest based on the activations of the neural network model.

Through concept probing, we can investigate whether the contents of a neural network's representations relate to the semantics of their respective concepts of interest. The central assumption is that high probe performance indicates that the probed representations encode the concept of interest. Probing is thus an essential step, enabling better interpretability and

helping to bridge the gap between subsymbolic representations and human-understandable symbolic concepts – one of the key steps in the neuro-symbolic cycle (Mileo, 2025).

There has been considerable work on concept probing, focusing primarily on what it reveals about the model being probed (Pimentel et al., 2020b; Alain and Bengio, 2017), or on the architecture and training of the probes (Sanh and Rush, 2021; Zhou and Srikumar, 2021; Pimentel et al., 2020a). Other works focus on exploring the application of concept probing to specific types of neural network models (Hupkes et al., 2018; Linzen et al., 2016), or to models performing tasks in specific domains, like game playing (Pálsson and Björnsson, 2024) and natural language processing (Tenney et al., 2019). Concept probing has also inspired the development of new methods to interpret neural network models: Ferreira et al. (2022) uses probe's outputs to induce theories describing a model's internal classification process; Lovering and Pavlick (2022) utilizes probes to assess whether a neural network's representations are consistent with a logic theory; Tucker et al. (2021) leverages probes to generate counterfactual behavior in neural networks; and de Sousa Ribeiro and Leite (2021) employs probes to produce ontology-based symbolic justifications for a neural network's outputs. All of the aforementioned works either consider the performance of the concept probes to make inferences regarding the model being examined or leverage the probes' outputs to perform some subsequent downstream task.

It turns out that throughout a model's layers, its representations change significantly, and thus, the performance of a probe is highly dependent on the specific representations considered. Some representations may allow for a concept to be linearly mapped, while others may require a highly complex, non-linear mapping, or may not encode the concept at all. It is thus essential to be able to identify which representations from a given model should be considered when developing a probe for some concept of interest. Concept probing typically focuses on representations resulting from a model's layer (Belinkov, 2022), as it provides a feasible and practical compromise between analyzing single units – which overlooks unit interaction – and pinpointing sets of units – which leads to an intractable search space. However, despite the significance of a probe's performance for concept probing, little attention has been given to this topic. Most work on concept probing focuses either on the model being probed or on the concept probes themselves, with current approaches often selecting an arbitrary layer to probe (Belinkov, 2022).

In this paper, we propose an efficient method to identify which layer's representations should be used when probing for a given human-defined concept of interest. Our approach is based on two main characteristics of the representations that are fundamental for the development of accurate probes: - how much information about the concept of interest is present in the representations; and - how regular are the representations regarding the concept of interest. The first characteristic tells us whether the concept is represented, and the second indicates how easily it can be probed for. We base our proposed method on information theory, which provides a formal framework for assessing these characteristics and practical approaches for estimating them. To validate our method, we consider various neural network models and datasets, showing that it efficiently identifies representations that enable the development of simpler and highly accurate concept probes. We discuss how the characteristics used by the method vary throughout a model's layers, and what might be inferred from them about the concept's representations. We conclude that training on

highly informative and regular representations enables the development of highly performant concept probes, even with limited training data.

The paper is organized as follows: Section 2 provides an overview of concept probing and Section 3 describes our method for characterizing the concept's representations. Section 4 presents the experimental setup and discusses how our characterization of the concepts varies throughout the layers of different neural network models, with Section 5 evaluating the probes trained based on the selected representations. We discuss some related work in Section 6 and conclude by summarizing our main findings in Section 7.

## 2. Concept Probing

Concept probing relies on the premise that neural networks distill useful representations layer by layer. Throughout the model, these representations gradually abstract away from the input space, moving towards representations that can be used to directly achieve the model's expected outputs. Concept probing leverages such representations by training a model – the probe – that observes the activations produced by some layer of a model and predicts a given concept of interest – also referred to as *property* (Belinkov, 2022). For example, consider a convolutional neural network trained to classify bird images. One might train a probe to identify whether a concept such as *having a needle-shaped bill* is detected from the activations of a layer of this model. The probe's performance is often used to assess how well these representations encode the concept of interest (Alain and Bengio, 2017).

More formally, let $f \colon x \mapsto y$ be a neural network model – often referred to as the *original model* – that maps input $x$ to output $y$, and which generates intermediate representations of $x$ in each of its layers $l$ – denoted by $f_l(x)$; $C$ the set of possible values of concept of interest $\mathsf{C}$; and $\mathcal{D} = \{(x_1, c_1), \ldots, (x_n, c_n)\}$ a dataset composed of pairs of input samples $x$ and values from $C$. A probe at layer $l$ of $f$ for $\mathsf{C}$ is a model $g \colon f_l(x) \mapsto c$, where $c \in C$. The dataset to train a probe at layer $l$ of $f$ for $\mathsf{C}$ given $\mathcal{D}$ is $\mathcal{D}_l = \{(f_l(x_1), c_1), \ldots, (f_l(x_n), c_n)\}$. Note that the semantics of the concept of interest is given extensionally by the dataset $\mathcal{D}$.

The performance of a probe $g$ at layer $l$ is measured on a separate test dataset $\mathcal{D}'_l$ constructed as $\mathcal{D}_l$ but with fresh instances. As these datasets are typically balanced regarding the concept values, the accuracy of $g$ on $\mathcal{D}'_l$ is generally considered. Our goal is to identify layers $l$ whose representations allow for the development of highly accurate probes $g$.

## 3. A Method for Selecting the Layer for Probing

In this section, we describe our method for characterizing a model's intermediate representations at each layer and selecting a layer to probe for a given concept of interest.

**Concept Informative Representations**  In order to train a probe based on the intermediate representations at some layer $l$ to predict a concept of interest $\mathsf{C}$, these representations must provide some information regarding the concept's values. In other words, given a dataset $\mathcal{D} = \{(x_1, c_1), \ldots, (x_n, c_n)\}$, observing $f_l(x_i)$ should reduce the uncertainty regarding the concept's value. This is captured by the notion of *mutual information*. With $f_l(\boldsymbol{x}) = (f_l(x_1), \ldots, f_l(x_n))$ and $\boldsymbol{c} = (c_1, \ldots, c_n)$, the mutual information of the intermediate representation at layer $l$ and the concept of interest $\mathsf{C}$ can be expressed as $I(f_l(\boldsymbol{x}); \boldsymbol{c}) = H(\boldsymbol{c}) - H(\boldsymbol{c}|f_l(\boldsymbol{x}))$, where $H(\boldsymbol{c})$ denotes the entropy of the concept's values,

and $H(\boldsymbol{c}|f_l(\boldsymbol{x}))$ the conditional entropy of those values given the representations. To facilitate the comparison between the resulting values obtained with different representations and concepts of interest, we use the *uncertainty coefficient*, given by $U(\boldsymbol{c}|f_l(\boldsymbol{x})) = \frac{I(f_l(\boldsymbol{x});\boldsymbol{c})}{H(\boldsymbol{c})}$,[1] which is a normalized version of the mutual information, describing the fraction of information that one variable provides regarding another one. For each layer of a model, we characterize how informative it is regarding some concept of interest by computing its uncertainty coefficient.

As mutual information captures all dependence between two random variables – and not just linear dependence – a higher uncertainty coefficient indicates the possibility of better predicting a concept given a layer's representations. However, a high uncertainty coefficient is not enough to guarantee the training of an accurate probe.

**Concept Regular Representations**   For a probe to train with limited data and still generalize, the underlying layer's representations should exhibit regularities – i.e., clear structure – with regard to the concept labels. Generally, the *simpler* these regularities are, the less data is required for the probe to identify them and properly generalize (Voita and Titov, 2020). The existence of clear regularities also allows for simpler probe models to be trained. The minimum description length principle (Rissanen, 1978) provides a framework for quantifying the complexity of a dataset relative to its associated labels. Given a dataset $\mathcal{D}$, the Shannon's coding theorem (Shannon, 1948) provides an optimal bound on the description length given by $-\sum_i \log_2 p(c_i|f_l(x_i))$, assuming the samples are independent and come from a probability distribution $p(\boldsymbol{c}|f_l(\boldsymbol{x}))$. In this way, one can estimate how regular the representations of a layer are wrt. a concept of interest. To estimate $p(c_i|f_l(x_i))$, we consider the probabilities given by a logistic regression classifier, providing an estimate of how well a simple probe encodes the layer's representations. This estimate corresponds to the categorical cross-entropy loss evaluated on this classifier. However, this estimate is unbounded and not generally comparable between datasets, so we consider a related quantity – the accuracy of the logistic regression classifier. For each layer of a model, we characterize how regular the representation is by estimating the accuracy of a logistic regression classifier trained on a dataset $\mathcal{D}$ using 5-fold cross-validation – we denote this value as $R(\boldsymbol{c}|f_l(\boldsymbol{x}))$.

In contrast to the informativeness of a layer's representations, highly regular representations ensure that accurate probes can be trained based on a layer's representations. However, low regularity does not imply that an accurate probe cannot be trained based on those layers' representations. Thus, to probe a given concept of interest, one should identify layers whose representations are both informative and regular. They should have sufficient information to predict a concept, while also allowing for a simple and direct mapping of the concept.

**Selecting a Model's Layer**   Given an original model $f$ and a dataset $\mathcal{D}$, our method consists of selecting the layer $l^*$ of $f$ such that:

$$l^* = \arg\max_l \ \lambda \ U(\boldsymbol{c}|f_l(\boldsymbol{x})) + (1 - \lambda)\frac{k \ R(\boldsymbol{c}|f_l(\boldsymbol{x})) - 1}{k - 1} \tag{1}$$

---

1. Note that, to compute the mutual information of an intermediate representation and some concept of interest, one would need to know their marginal and joint distributions. However, such distributions are, in practice, unknown. Given the formal limitations on measuring mutual information (McAllester and Stratos, 2020), throughout this paper, we estimate this quantity using the method from (Noshad et al., 2019), designed to estimate the mutual information of high-dimensional multivariate random variables – which aligns with our requirements as a layer's representations may have a very high dimensionality.

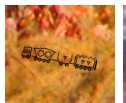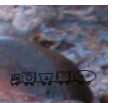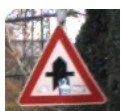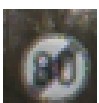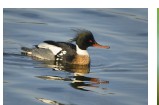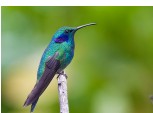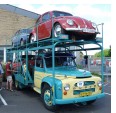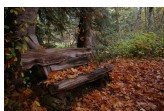

Figure 1: Sample images from XTRAINS, GTSRB, CUB, and ImageNet datasets.

where $k$ is the cardinality of the concept of interest (e.g., $k = 2$ for a binary concept), and $\lambda \in [0, 1]$ defines the relative importance between information and regularity. If $\lambda = 0$ (resp. $\lambda = 1$), only the regularity (resp. information) of the representation is accounted for.

## 4. Tracking a Concept's Representation Throughout a Model's Layers

In this section, we first introduce four image classification datasets and six neural network models used to assess our method, and then discuss how the representations of various concepts of interest vary throughout the model's layers. The datasets were selected to represent different scenarios, encompassing concepts of varying levels of abstraction and complexity. Figure 1 shows sample images from each dataset. When assessing how informative and regular a layer's representation is, a balanced dataset $\mathcal{D}$ of at most 1 000 samples was considered (on average, 770 were used, as some concepts had a limited number of samples available).

**Explainable Abstract Trains Dataset (XTRAINS)** (de Sousa Ribeiro et al., 2020): synthetic dataset of trains on diverse backgrounds. Meant for benchmarking explainability methods, it contains an ontology describing how the labeled concepts relate to each other. Three types of trains (TypeA, TypeB, and TypeC) are defined based on their visual characteristics. We probe three VGGNet models (Simonyan and Zisserman, 2015) from (Ferreira et al., 2022) – referred to as $f_A$, $f_B$, and $f_C$[2] – trained to identify trains of the respective type, each achieving an accuracy of about 99% on a balanced test set of 10 000 images.

**German Traffic Sign Recognition Benchmark (GTSRB)** (Stallkamp et al., 2011): dataset with images of 43 types of traffic signs. We also consider the ontology and labels from (de Sousa Ribeiro et al., 2025a), which are based on the 1968 Convention on Road Signs and Signals (United Nations, 1968) and describe each type of traffic sign based on visual concepts. E.g., a stop sign is described as having an octagonal shape, a red ground color, and a white 'stop' symbol. As original model, we probe a MobileNetV2 (Sandler et al., 2018) – which we refer to as $f_{GTSRB}$ – trained to identify each type of traffic sign and having an accuracy of 98% on the dataset's test set.

**Caltech-UCSD Birds-200-2011 (CUB)** (Wah et al., 2011): this dataset is composed of images of birds from 200 species. Each image is labeled with various additional attributes representing visual concepts that are described to be relevant for the identification of the bird species. As original model, we consider the ResNet50 from (Taesiri et al., 2022), pre-trained in the iNaturalist dataset (Horn et al., 2018) and fine-tuned in CUB, achieving an accuracy of about 86% on the dataset's test data. We refer to this model as $f_{CUB}$. As reported e.g. in (Zhao et al., 2019; Koh et al., 2020), some of the attributes in CUB were noisily labeled. For this reason, we consider the revised labels from (de Sousa Ribeiro et al., 2025a).

---

2. Details regarding the original models being probed can be found in (de Sousa Ribeiro et al., 2025b).

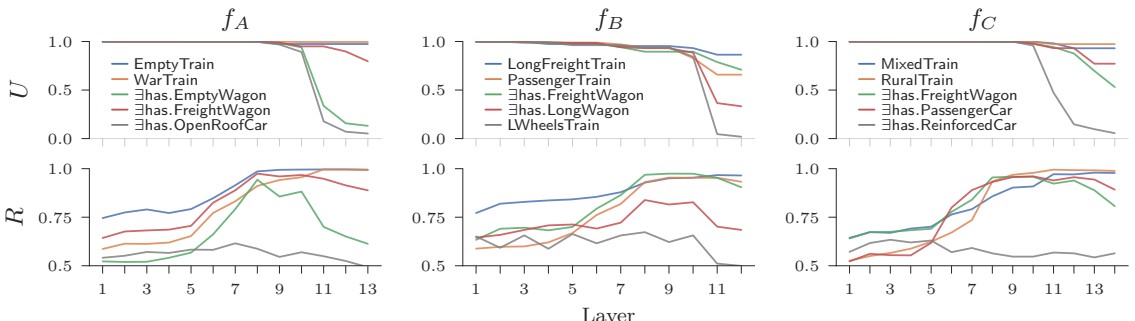

Figure 2: Characterization throughout a model's layers (XTRAINS).

**ImageNet Object Attributes (ImageNet)** (Russakovsky and Fei-Fei, 2010): this dataset contains images from 384 ImageNet synsets labeled with 25 concepts regarding the visual characteristics of the objects in the images. For example, the concept of Red indicates whether the image contains an object that is at least 25% red. As original model, we probe the ResNet50 (He et al., 2016) model from (PyTorch Foundation, 2025) achieving an accuracy of about 81% on the ImageNet-1K test set (Russakovsky et al., 2015), which we refer to as $f_{ImageNet}$.

**Probed Concepts** For each of the six original models, we probe five random concepts of interest. For the original models trained in XTRAINS and GTSRB, we used their ontologies to ensure that one of the probed concepts was not related to the task of the original model. This allows one to assess if the proposed characterization of a model's representations differentiates such concepts. Additionally, to contrast with the more abstract high-level concepts that were labeled in the ImageNet dataset, we considered the concept of Reddish, characterizing images with at least 10% of its pixels being *red pixels* i.e., when the difference between their red component and the mean of the blue and green components is high ($> 150$).

**Results and Discussion** The characterization for how informative and regular the intermediate representation of each concept are, at the different layers of a model, is shown in Figure 2, for $f_A$, $f_B$, and $f_C$, and in Figure 3 for $f_{GTSRB}$, $f_{CUB}$, and $f_{ImageNet}$.

Our first observation is that the proposed characterization is capable of distinguishing concepts that are not related to the task of its original model. This is evidenced by how the representations of each of these concepts - shown in gray color – are easily recognizable, having the lowest regularity throughout each respective model and being the first to have their information discarded in each model.

Our second observation is that the proposed characterization is capable of distinguishing between concepts with different characteristics, identifying where in a model they are more amenable to be probed from. This is supported by how different sets of concepts are characterized throughout each original model. For $f_A$, $f_B$, and $f_C$, we observe that the representations for concepts related to individual wagons of a train – e.g., ∃has.FreightWagon – achieve their highest regularity before concepts related to the whole trains – e.g., WarTrain. This is particularly interesting, given that the concepts related to whole trains are often defined based on those regarding individual wagons. It suggests that this characterization is able to capture that the models first encode the simpler wagon-related concepts and then leverage those representations to detect the more complex train-related concepts, at which

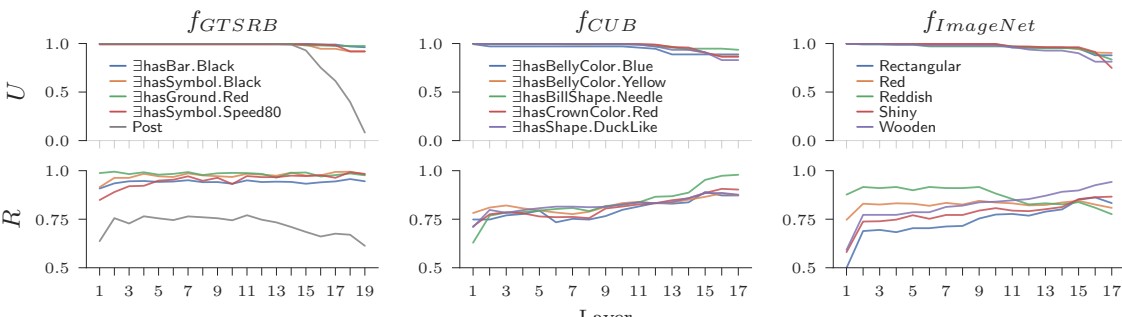

Figure 3: Characterization throughout a model's layers (GTSRB, CUB, ImageNet).

point they started to discard the information regarding the wagon-related concepts. For $f_{CUB}$, we observe that the information regarding the probed concepts seems to remain high until the latter layers of the model, while the regularity of the representations seems to steadily increase throughout the model. This reflects the nature of these concepts, which are rather high-level, representing specific visual attributes that experts use to classify different bird species. Similarly, in $f_{ImageNet}$ the information regarding the probed concepts seems to remain high until the latter layers of the model. The regularity of the concepts' representations seems to generally increase throughout the model, with the exception of the more concrete low-level Reddish concept, which decreases throughout the model.

Our third observation is that the proposed characterization does not only provide useful information for probing, but also regarding the original model and how it might be revised. For $f_{GTSRB}$, the probed concepts are generally simple – relating to colors and shapes. This is reflected in the results, where both the information and regularity of the representations of the four relevant concepts remain fairly high and constant throughout the model's layers. This suggests that a simpler model could have been considered for this classification task.

These results provide interesting insights regarding how the concepts of interest are encoded in an original model – some concepts are low-level and encoded in the first layers ($\exists$hasGround.Red in $f_{GTSRB}$); others are high-level and gradually develop throughout the model (e.g., $\exists$hasBillShape.Needle in $f_{CUB}$); some concepts seem to be a stepping stone towards more abstract higher-level concepts (e.g., $\exists$has.EmptyWagon in $f_A$); and others seem not to be encoded at all (e.g., $\exists$has.ReinforcedCar in $f_C$). In general, these characteristics – information and regularity – seem relevant for identifying where and how a given concept of interest is represented in a model. This allows for an informed decision regarding which layer should be considered when probing for this concept.

## 5. Empirical Evaluation of the Selected Layer

To estimate the quality of the layer selected by the proposed method for each concept of interest, we train and test a probe. The choice of the architecture for probing models is a debated issue, with some arguing for the use of simpler probing models (Alain and Bengio, 2017; Liu et al., 2019), while others argue for more complex ones (Pimentel et al., 2020a,b; Tucker et al., 2021). To cater to the different sides of this debate, we consider a variety of probes $g$: a logistic regression classifier, a ridge classifier, a LightGBM decision tree (Ke et al., 2017), a neural network, and a mapping network (de Sousa Ribeiro and Leite, 2021).

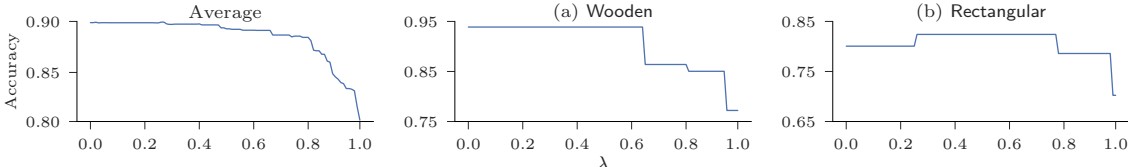

Figure 4: Ablation of $\lambda$ value over all concepts and models, and for two specific concepts.

| Model | Our Method | | Layers' Avg. | Oracle | % Oracle | Best Validation | | Input Reduce | |
|---|---|---|---|---|---|---|---|---|---|
| | Acc. | Time | Acc. | Acc. | | Acc. | Time | Acc. | Time |
| $f_A$ | 89.3 | **9.6** | 70.6 | 89.7 | 99.6 | **89.4** | 276.3 | 86.5 | 114.7 |
| $f_B$ | **85.6** | **9.2** | 71.1 | 87.8 | 97.3 | 85.4 | 228.0 | 82.8 | 101.1 |
| $f_C$ | **89.4** | **9.5** | 71.2 | 90.3 | 99.0 | 88.9 | 290.5 | 88.5 | 161.3 |
| $f_{GTSRB}$ | **93.9** | **3.1** | 90.7 | 95.0 | 98.7 | 92.8 | 315.8 | 90.7 | 75.1 |
| $f_{CUB}$ | **94.8** | **11.8** | 82.5 | 95.8 | 98.9 | 94.3 | 487.7 | 94.0 | 175.4 |
| $f_{ImageNet}$ | **88.0** | **10.8** | 79.8 | 89.5 | 98.4 | 87.6 | 408.3 | 82.8 | 108.7 |
| Average | **90.2** | **9.0** | 77.7 | 91.4 | 98.6 | 89.7 | 334.4 | 87.6 | 122.7 |

Table 1: Average probe accuracy (%) and method runtime (min) for each dataset.

**The Probes**  Each probe is trained using a balanced dataset $\mathcal{D}$ of at most 1 000 samples, with 20% of the training data being used for validation purposes. For a given concept of interest and layer, we report the accuracy of the probe model with the highest validation accuracy on a separate test dataset with a similar size to the training set.

For the ridge classifier, we perform a hyperparameter search over the alpha values of [0.01, 0.05, 0.1, 0.5, 1, 5, 10, 50, 100]. The LightGBM probe is used with default parameters, and a validation set is used together with early stopping to select the number of boosting rounds. The neural network probe has a feedforward architecture with ReLU non-linearity and a hidden layer of size 10. The mapping network probe shares the same architecture, but L1 regularization is applied to its weights with a strength of 0.001. Mapping network probes are trained using the input reduce procedure described in de Sousa Ribeiro and Leite (2021) to further reduce a layer's representation. Early stopping with a patience of 15 is used to select the number of training epochs for both neural network and mapping network probes.

**Results and Discussion**  We begin by evaluating the effect of $\lambda$ (from Equation 1) in our method. Figure 4 shows the average accuracy of the resulting probes for the selected layer over all concepts and models, while varying the $\lambda$ value. The resulting accuracy seems to be relatively stable for $\lambda$ values lower than 0.6. Higher $\lambda$ values, which mostly neglect how regular the layers' representation were, generally lead to worse results. Additionally, we show the results for the Wooden and Rectangular concepts in $f_{ImageNet}$. These illustrate concepts that are sensitive to the tuning of the $\lambda$ value, probed from the same model, but having distinct optimal ranges. Nevertheless, the cost of fine-tuning $\lambda$ is low – for a given concept, it leads to considering only, on average, 3.7 different layers. In the remainder, we consider $\lambda = 0.26$, the value that results in the best average performance.

Table 1 shows, under 'Our Method', the average test accuracy of the probes trained using the representations of the layer selected by our method for each original model, and the method's runtime. We first compare our method with a relevant baseline – computed by averaging the performance of the probes across all layers of the model – shown in column 'Layer's Average'. This comparison is quite relevant, as other works often arbitrarily select

the layer used for probing. Our method vastly outperforms this baseline, indicating that it selects layers that enable much more accurate probe training than if one were to make an uninformed guess about which layer's representations should be considered. We also compare our results against an upper bound resulting from an *oracle* that would always select the layer that resulted in the probe with the highest *test accuracy* – shown in column 'Oracle'. Note that this method should not be regarded as a selection method, as its results would be biased due to considering the test accuracy. Column '% Oracle' shows our method's results as a percentage of the oracle's, indicating that they are quite close to these *"ideal"* results.[3]

We also compare the results produced by our method to those of two other existing methods. The first method is an exhaustive search procedure to select which layer to probe, training probes for all layers, and selecting the one with the highest validation accuracy. This is shown in Table 1 under 'Best Validation'. We found that our method slightly outperforms this approach, likely due to some overfitting introduced by the selection based on the validation accuracy. This result is quite encouraging since the approach based on the validation accuracy is generally unfeasible, as it requires the training of many models. This is illustrated by the runtimes shown in Table 1.[4] Additionally, we verified that the layers selected by our method allow for the use of simpler probing models: whereas our approach led to the use of logistic regression and ridge probes more often, considering the validation accuracy led more often to the use of LightGBM and neural network probes. We also compare our method to the *input reduce* procedure from (de Sousa Ribeiro and Leite, 2021) – shown under 'Input Reduce'. This procedure iterates the layers of a model, starting from the last, to pinpoint the particular units that should be considered when probing for a concept, rather than selecting a layer. Our method produced superior results, which we attribute to the *input reduce* procedure stopping its search for units once it reaches a layer where no units are selected, thus missing out on some important internal representations.

These results support the claim that our method for characterizing layer representations facilitates the efficient identification of layers that lead to accurate concept probes.

## 6. Related Work

**Interpretability and the Need for Human-Defined Concepts** The growing use of neural network models across diverse fields has driven the development of various methods to enhance their interpretability and explainability. Early approaches were typically proxy-based (Ribeiro et al., 2016; Augasta and Kathirvalavakumar, 2012; Schmitz et al., 1999), replacing neural networks with interpretable models that mimic their input-output behavior, or relied on saliencies and attributions (Ivanovs et al., 2021; Rebuffi et al., 2020; Sundararajan et al., 2017), assigning importance scores to input features to explain predictions.

Although these methods offered some insight into model behavior, user studies reveal that their explanations were often unhelpful or ignored by end users (Adebayo et al., 2020; Chu et al., 2020; Shen and Huang, 2020), mainly because such methods explain models in terms of input features, which may lack symbolic meaning or fail to align with users' understanding. For instance, raw image pixels hold little standalone meaning, so attributing importance to specific pixels can be uninformative if users cannot interpret their meaning.

---

3. Results for individual concepts can be found in (de Sousa Ribeiro et al., 2025b).

4. Details regarding the computational resources can be found in (de Sousa Ribeiro et al., 2025b).

The need for symbolically meaningful explanations has given rise to Concept-based Explainable AI (Poeta et al., 2023), which addresses the shortcomings of earlier methods by allowing models to be interpreted through human-defined concepts. This includes techniques for identifying latent concepts (Räuker et al., 2023), concept probing (Belinkov, 2022), and explaining model outputs via human-defined concepts (Michel-Delétie and Sarker, 2024).

**Representation Identification** Understanding what is encoded in neural network models has attracted significant interest. Some have used visualization techniques to interpret individual units (Goh et al., 2021; Nguyen et al., 2016), while others have taken more formal approaches (Dalal et al., 2024; Dalal, 2024; Mu and Andreas, 2020). Efforts also include identifying concepts in the representations of a single layer (Ghorbani et al., 2019) or across all layers (Horta et al., 2021). These methods help identify which concepts are present in a model, addressing a key assumption of concept probing – that the concepts to be probed are known in advance. While unit-focused methods like Network Dissection (Zhou et al., 2019) and CLIP Dissect (Oikarinen and Weng, 2023) clarified the role of individual units, concept probing targets human-defined concepts the model was not explicitly trained for, which may not align with specific units. Others have studied how the representations of the output concepts of a model evolve across its layers (Noshad et al., 2019; Alain and Bengio, 2017). In contrast, we consider concepts other than the model's output.

**Relation to Information-Theory** Others have drawn connections between concept probing and information theory. Pimentel et al. (2020b) operationalize concept probing as estimating the conditional mutual information of some concept of interest given the representations, with higher-performing probes indicating that the representations carry more information about the concept. Shwartz-Ziv and Tishby (2017) studies the internals of neural network models by examining how the mutual information of representations wrt. the input and wrt. the output of a model varies throughout its layers. Voita and Titov (2020) uses the minimum description length to inform the design of the concept probes.

## 7. Conclusions

In this paper, we proposed a method for efficiently selecting a layer from a neural network model whose representations allow for the accurate probing of a given human-defined concept of interest. The key insight lies in characterizing each layer's representations based on how informative and regular they are wrt. the concept being probed. We support the assessment of these characteristics by considering an information-theoretic approach. We showed that the resulting probes developed based on the selected layer's representations are highly performant, achieving higher accuracy than those obtained from existing methods.

We also found that observing how these characteristics vary throughout a model's layers provides relevant insights regarding the nature of the probed concepts and how they are encoded in the model, which may further inform the design of the probing model.

We conclude that knowing how informative and regular the representations of a model are allows one to make an informed decision regarding which layer of a model should be considered when probing a given concept of interest. We believe this work makes a valuable contribution to allowing for a more streamlined development of accurate concept probes. This is especially critical in neuro-symbolic frameworks, where the efficacy of these probes directly impacts the performance of downstream tasks.

## Acknowledgments

This work was supported by FCT I.P. through UID/04516/NOVA Laboratory for Computer Science and Informatics (NOVA LINCS) and through PhD grant (DOI 10.54499/UI/BD/ 151266/2021), and by Project Sustainable Stone by Portugal - Valorization of Natural Stone for a digital, sustainable and qualified future, nº 40, proposal C644943391-00000051, co-financed by PRR - Recovery and Resilience Plan of the European Union (Next Generation EU).

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
