# OpenReview forum: "Concept Probing: Where to Find Human-Defined Concepts"
_nesyconf.org/NeSy/2025/Conference_Phase_2 — NeSy 2025 - Phase 2 Poster_

### Official Review · Reviewer_7fH9 · 2025-07-04
**Paper improved the last version**

**Rating:** 7
**Confidence:** 3

**Review:**

The paper proposes a method to identify which layer in a neural network should be considered when probing a human-defined concept. In this version, the authors provided more comparison between other works and methods. Their method uses the balance between the information and regularity for a specific concept. I would include some dataset images to describe the concepts probed. Following some comments on the text:
- Remove the § from the text
- "The mapping network probe shares the same architecture, but L1 regularization is applied to its weights with a strength of 0.001.": Why does it matter for the process?
- "We begin by evaluating the effect of λ in our method.": What does the lambda represent? It was cited in the formula, but at this moment we need to be reminded.
- "we consider λ = 0.26, the value that results in the best average performance.": What does this value mean?
- "considering only 3.7 different layers": How can you consider 3.7 layers? It should be an integer number, shouldn't it?

**Anonymity:**

Remain anonymous

---

### Official Review · Reviewer_cP7L · 2025-07-06
**Unclear fit to NeSy**

**Rating:** 4
**Confidence:** 4

**Review:**

The paper proposes a method to identify which layer’s representations in a neural network model should be considered when probing for a given human-defined concept of interest. Overall the paper is well writting, though it should better explain why this is a good concept for NeSy. Without this, almost every machine learning problem would fit nesy. For instance, what ar the concepts you consider? Are they logical formulas? Are they jsut a finite set of class labels? This is also related to the meaning of "∃hasGround.Red", "∃hasBillShape.Needle" and Co which are not really introduced. So overall, while interesting, the paper in its current form does not match well the idea of neurosymbolic. Please not that indeed it might not be much work to get this done but right now it seems rather an information theory or just ML paper.

**Anonymity:**

Remain anonymous

---

### Official Review · Reviewer_GK6S · 2025-07-06
**Revised submission significantly improved**

**Rating:** 6
**Confidence:** 4

**Review:**

While I still think that Nesy may not be a strong fit for this type of paper, and that the novelty is marginal with respect to existing methods, the paper was substantially strenghtened with respect to the first round. In particular, I found the comparison with existing baselines and ablation studies thorough and convincing of the strenghts of the method in terms of quality of the extracted concept mapping and time needed to compute it.

**Anonymity:**

Remain anonymous

---

### Official Review · Reviewer_JZ4a · 2025-07-09
**I appreciate the clarifications and changes**

**Rating:** 7
**Confidence:** 5

**Review:**

As per the title.  I now understand what "best validation" means :-)  This is enough for me.

**Anonymity:**

Remain anonymous